# Characterisation of Aberrant Metabolic Pathways in Hepatoblastoma Using Liquid Chromatography and Tandem Mass Spectrometry (LC-MS/MS)

**DOI:** 10.3390/cancers15215182

**Published:** 2023-10-28

**Authors:** Alison Whitby, Pardeep Pabla, Bhoomi Shastri, Laudina Amugi, Álvaro Del Río-Álvarez, Dong-Hyun Kim, Laura Royo, Carolina Armengol, Madhumita Dandapani

**Affiliations:** 1Children’s Brain Tumour Research Centre, School of Medicine, Biodiscovery Institute, University of Nottingham, Nottingham NG7 2RD, UK; mszaw2@exmail.nottingham.ac.uk (A.W.); mszbs1@exmail.nottingham.ac.uk (B.S.); 2School of Medicine, Royal Derby Hospital Centre, University of Nottingham, Derby DE22 3DT, UK; mbzpp5@exmail.nottingham.ac.uk; 3Centre for Analytical Bioscience, Advanced Materials and Healthcare Division, School of Pharmacy, University of Nottingham, Nottingham NG7 2RD, UK; l.k.amugi@bham.ac.uk (L.A.); dong-hyun.kim@nottingham.ac.uk (D.-H.K.); 4Phenome Centre Birmingham, School of Biosciences, University of Birmingham, Birmingham, B15 2TT, UK; 5Childhood Liver Oncology Group, Translational Program in Cancer Research (CARE), Germans Trias i Pujol Research Institute (IGTP), 08916 Badalona, Spain; adelrio@igtp.cat (Á.D.R.-Á.); lroyo@igtp.cat (L.R.); carmengol@igtp.cat (C.A.); 6Centro de Investigación Biomédica en Red (CIBER) en Enfermedades Hepáticas y Digestivas, 28029 Madrid, Spain

**Keywords:** hepatoblastoma, liquid chromatography and tandem mass spectrometry, metabolomics, acylcarnitine, fatty acid oxidation, carnitine palmitoyl transferase (CPT1)

## Abstract

**Simple Summary:**

Hepatoblastoma is a rare childhood liver cancer with poor outcomes for high-risk patients. Better treatments and better ways of identifying patients who respond poorly to treatment are needed. This paper uses new methods for identifying chemicals or metabolites produced in the tumour. By comparing the profiles of these metabolites in tumour tissue versus normal liver tissue taken from the same patient, we demonstrated that some metabolites differ significantly in hepatoblastoma. This correlates with gene expression data, suggesting that we identified the metabolites correctly. We also stained tumour tissues for proteins (enzymes) that regulate transport of fatty acids into the mitochondria, which are the cell’s powerhouses. Taken together, our results indicate that tumour cells change the energy sources they use and rewire the cellular systems accordingly. Further work is required to verify this, but these leads could improve our understanding of the disease and lead to the development of novel therapies.

**Abstract:**

Hepatoblastoma (HB) is a rare childhood tumour with an evolving molecular landscape. We present the first comprehensive metabolomic analysis using untargeted and targeted liquid chromatography coupled to high-resolution tandem mass spectrometry (LC-MS/MS) of paired tumour and non-tumour surgical samples in HB patients (*n* = 8 pairs). This study demonstrates that the metabolomic landscape of HB is distinct from that of non-tumour (NT) liver tissue, with 35 differentially abundant metabolites mapping onto pathways such as fatty acid transport, glycolysis, the tricarboxylic acid (TCA) cycle, branched-chain amino acid degradation and glutathione synthesis. Targeted metabolomics demonstrated reduced short-chain acylcarnitines and a relative accumulation of branched-chain amino acids. Medium- and long-chain acylcarnitines in HB were similar to those in NT. The metabolomic changes reported are consistent with previously reported transcriptomic data from tumour and non-tumour samples (49 out of 54 targets) as well as metabolomic data obtained using other techniques. Gene set enrichment analysis (GSEA) from RNAseq data (*n* = 32 paired HB and NT samples) demonstrated a downregulation of the carnitine metabolome and immunohistochemistry showed a reduction in CPT1a (*n* = 15 pairs), which transports fatty acids into the mitochondria, suggesting a lack of utilisation of long-chain fatty acids in HB. Thus, our findings suggest a reduced metabolic flux in HB which is corroborated at the gene expression and protein levels. Further work could yield novel insights and new therapeutic targets.

## 1. Introduction

Paediatric liver cancer is rare with increasing incidence over the past few decades, from around 1–1.5 per million to over 2 per million [1,2]. The commonest tumour seen usually in children under 5 years of age is hepatoblastoma (HB). Increased incidence of HB is associated with prematurity or low birth weight and familial conditions, such as Beckwith–Weidemann syndrome (BWS) and familial adenomatous polyposis (FAP) [3]. Hepatocellular carcinoma (HCC), much rarer, is usually seen in the teenage population and is usually associated with underlying cirrhosis, secondary to perinatal hepatitis B infection or metabolic diseases [4]. Current treatment of paediatric HB is based on clinical risk stratification as, until recently, most tumours were diagnosed based on radiology and elevation of the tumour marker Alpha fetoprotein (AFP), with histology only being analysed following resection. Whilst most patients can be cured with a combination of chemotherapy and surgery, those with metastatic disease still have poor outcomes.

Histological and biological advances over the recent decades have identified that these tumours can have mixed overlapping features, and it is sometimes challenging to distinguish HB from HCC; in recent years, a provisional entity with clinical and histopathological features of HB and HCC called HCN-NOS (hepatocellular neoplasm, not otherwise specified) has been coined [5]. The current clinical trial PHITT (Paediatric Hepatic International Tumour Trial), enrolling patients across three large international cooperative groups, the Children’s Oncology Group (COG), the SIOP Epithelial Liver (SIOPEL) group across Europe and the Japanese Children’s Cancer Group (JCCG), aims to collect diagnostic and surgical histological samples to prospectively annotate and validate the biology of these rare tumours. This trial uses the CHIC (Children’s Hepatic tumour International Collaboration) stratification to identify four distinct risk groups of HB [6]. Whilst the impact of the clinical risk factors is clear, there is a need to identify prognostic and therapeutic biomarkers that can help define high-risk disease as well as aid the development or selection of novel therapies for these patients.

It is well known that therapy response in cancers, including paediatric cancers, is associated with their biology. Several studies have explored genetic and transcriptomic profiles in search of the genomic hallmarks of HB pathogenesis. Early genetic studies revealed that HB cells are often diploid (foetal type) or hyperdiploid, with a limited number of chromosomal abnormalities. Recurrent chromosomal alterations involve chromosomes 2, 20, 1 and 8 [7]. The most frequent mutated gene is CTNNB1, coding for β-catenin; other less frequent mutated genes include ARID1, TERT promoter, APC and NFE2L2. There is growing evidence that prognostication can be improved by combining transcriptomic and epigenomic data with clinical stratification [8,9,10,11]. In 2008, two major molecular subtypes of HB, called C1/C2, and a 16-gene discriminating signature in tumours were described and the latter was shown to be an independent prognostic factor [9]. The prognostic impact of the 16-gene signature has since been validated in a multinational cohort of 174 HB patients [10]. Molecular risk stratification (MRS) of HB by integrating novel epigenomic biomarkers (i.e., 14q32-signature and epigenetic Epi-CB/A classification) has also been described in a cohort of patients, which improves CHIC stratification [11]. Epigenetic profiling has also revealed multiple epigenetic alterations, including hypomethylation of ASCL2 that regulates Wnt signalling as well as foetal-liver-like methylation patterns of IGF2 promoters [12,13].

Metabolomics is a rapidly growing field of post-genomic biology focusing on system-wide studies of metabolite levels and transformations in biological samples [14]. We know that cancer cells have dysregulated metabolism to promote rapid proliferation. The detection of these abnormal cancer metabolites through metabolomics could potentially lead to the discovery of tumour biomarkers and novel therapeutic targets. Two publications from the same group using high-resolution magic-angle-spinning nuclear magnetic resonance (NMR) spectroscopy (HR-MAS) metabolomic analysis of tumour and control liver pieces defined aberrant pathways in hepatoblastoma [15,16], namely, lipid metabolism, aerobic glycolysis and glutaminolysis. Interestingly, these studies showed that using tumour and non-tumour matched pairs from the same patient was important for finding biologically meaningful results, as glutamine and glutamate were increased and alanine increased in abundance in non-paired samples (which included some paired samples), whereas glutamine and glutamate were decreased yet alanine still increased in abundance in paired samples.

To date, there have been no metabolomics studies using the highly sensitive technique which can detect 100s of metabolites and give firm metabolite identifications and concentrations, namely, liquid chromatography with tandem mass spectrometry (LC-MS/MS). Untargeted metabolomics using LC-MS/MS gives a broader understanding of dysregulated metabolism, as it detects a greater number of related compounds within pathways, affected only by the extraction technique. This paper uses a simple metabolite extraction method with untargeted and targeted LC-MS/MS to identify and validate novel targetable metabolic alterations in HB in paired tumour and non-tumour liver tissue samples taken from the same patient.

## 2. Methods

### 2.1. Samples

Eight tumour and non-tumour matched-pair tissue samples were obtained from paediatric patients with a diagnosis of hepatoblastoma (HB) at the time of surgery and flash-frozen. For immunohistochemistry (IHC), additional formalin-fixed paraffin-embedded (FFPE) paired non-tumour and tumour samples from 15 patients with HB were also included in the study. All samples were collected in accordance with European and Spanish law. Informed written consent was obtained from each patient in accordance with European guidelines for biomedical research. The study conformed to the ethical guidelines of the 1975 Declaration of Helsinki, and it was approved by the Human Ethics Committee of the Hospital Universitari Germans Trias i Pujol. ISCIII National Biobank Registry, collection section, ref. C.0000226; samples were obtained as per biobank ethical approvals.

### 2.2. Untargeted Metabolomics

Liver samples (5 mg) were extracted using a homogeniser for 60 s in 200 µL 8:2 *v/v* methanol/water. The mixture was centrifuged at 10,000× *g* for 10 min and the supernatant was transferred to HPLC vials for LC-MS and LC-MS/MS analysis. Pooled QC was prepared post-extraction by transferring 20 μL of supernatant from each sample, vortexing and aliquoting into a HPLC vial. Methanol/water 8:2 (200 µL) was processed without liver, in the same way as the samples, as a reagent blank.

The analytical run followed this sequence of injections: a blank injection followed by 5 different mixtures of 268 standards (for identification) and randomised extracted samples (injected three times each) in a single batch. The column was conditioned with pooled QC and pooled QC injections were interspaced throughout the run to check the stability, robustness, repeatability and performance of the analytical system. The analytical method used was similar to that previously published, with the exception that the injection volume was 5 µL and the normalised collision energy (NCE) was stepped at 10, 20 and 40 [17,18].

Data processing, including metabolite identification, was performed by Compound Discoverer 3.3 (Thermo Fisher Scientific, Hemel Hempstead, UK) using a tailored untargeted metabolomics workflow (Appendix A). Metabolite identification was performed by matching accurate masses of the detected peaks with metabolites in BioCyc (human), the Human Metabolome Database and KEGG; the retention times (RTs) were obtained with 268 authentic standards (mass list for untargeted liver tumour and non-tumour.massList) and/or ddMS/MS with *mz*Cloud (HighChem HighRes identity search with an activation energy tolerance of 10) from a fragmentation database (Thermo Fisher Scientific, Hemel Hempstead, UK); and identification levels reported are according to the metabolomics standards initiative [17,19,20]: level 1, match of accurate mass, MS/MS fragmentation and retention time to authentic standard co-analysed with the samples under identical experimental conditions; level 2, match of accurate mass and retention time (two orthogonal data) to the authentic standard or match of accurate mass and MS/MS spectrum with compound in a library when data were taken under the same acquisition parameters; level 3, match of predicted retention times or predicted MS/MS spectra or both due to the lack of standards; level 4, unambiguously assigned molecular formulas where insufficient evidence exists to propose possible structures. Univariate analysis after log_10_ transformation (*t*-test with Benjamini–Hochberg false-discovery rate correction) was performed by Compound Discoverer and multivariate analysis (MVA) by Simca P+16 (Umetrics AB, Umea, Sweden), with imported datasets mean-centred and Pareto-scaled for MVA. The permutation test was performed with 200 permutations.

### 2.3. Targeted Metabolomics

Frozen liver samples were freeze-dried at −56 °C under sealed vacuum conditions to eliminate the effect of any changes in tissue water content on analyte concentrations. The dried sample was inspected for any visible blood before being weighed on a four-decimal-place balance. Acylcarnitines and amino acids were extracted using a modified version of the method described by Sun et al. [21]. Briefly, following the addition of 500 μL of isopropanol/1 M KH_2_PO_4_ buffer 1:1 (*v:v*) containing 50 μL of internal standard mixture (spiked into the extraction solvent and prepared as described below), dried liver samples were homogenised within the extraction solvent using a pestle and vigorously vortexed for 5 min. Then, following the addition of 500 μL of acetonitrile, samples were vigorously vortexed for a further 5 min and centrifuged for 20 min at 14,000× *g* at 4 °C. The supernatant was removed and evaporated under vacuum centrifuge at room temperature. Dried samples were resuspended in 100 μL of methanol: water 1:1 (*v:v*) and gently mixed for 5 min at 4 °C. Samples were centrifuged at 14,000× *g*, and the supernatant was removed and stored at −80 °C until analysis.

A range of acylcarnitine (C_2_-C_16_), free carnitine and the 20 naturally abundant amino acid (AAs) standards (Sigma-Aldrich, Gillingham, UK) were prepared as stock solutions at 1 mg/mL in methanol: water 1:1 (*v:v*) and diluted to create calibration standards across an appropriate concentration range for each analyte. Deuterated acylcarnitine internal standards, NSK-B, and deuterated and ^13^C branched-chain amino acids (BCAAs) internal standards, NSK-BCAA (CK isotopes, Newtown Unthank, UK), were prepared as per the manufacturer’s instructions and spiked into the extraction solvent as described above. Calibration standards were spiked into a proxy matrix, 7.5% bovine serum albumin (BSA), in phosphate-buffered saline (PBS) and extracted identically to the liver samples described above (separate standards for acylcarnitines and amino acids). This proxy matrix has been successfully used as a surrogate for liver samples previously [21]. Extracted standard concentrations vs. peak area ratio (of unlabelled standard relative to an isotopically labelled internal standard) were used to construct calibration curves and for quantification of free carnitine, acylcarnitines and BCAAs. AAs were quantified using extracted peak areas vs. concentration calibration curves. The analysis was performed on the same LC-MS as used for the untargeted work. Two separate injections of the same sample were used for reversed-phase C18 and HILIC analysis. The injection volume was 5 μL, and samples were maintained at 4 °C during the analysis. Acylcarnitines were separated using an ACE PFP–C18 column (100 × 2.1 mm, 2 µm pore size; Avantor, Theale, Reading, UK) as described previously [22]. The ZIC-pHILIC column was used for the separation of BCAAs and free carnitine similar to the untargeted analysis but with a longer gradient: starting with 20% (A), it increased to 95% (A) over 16 min, followed by equilibration to give a 24 min run time [23]. MS was performed in simultaneous ESI+ and ESI− full-scan modes with spray voltages of 3.5 (ESI+) and 2.5 kV (ESI−) and capillary voltages of 40 (ESI+) and −30 V (ESI−). In both modes, the sheath-, auxiliary- and sweep-gas flow 95 rates were 40, 11 and 2 arbitrary units, respectively, and the capillary and heater temperatures were 300 and 400 °C, respectively. Automated gain control (AGC) was targeted at 1 × 10^4^. The isolation width of the precursor ion was set at 0.7 (*m*/*z*). Mass resolution was set at 70,000 from *m*/*z* 100 to 600. Significant differences in metabolite levels between tumour and non-tumour samples were judged by a paired two-tailed t-test, with *p* < 0.05 considered significant.

### 2.4. Gene Set Enrichment Analysis

Gene set enrichment analysis (GSEA) was used to evaluate the correlation of specific gene lists downloaded from MSigDB with two different sample groups (phenotypes). Briefly, this method calculates an enrichment score after ranking all genes in the dataset based on their correlation with a chosen phenotype and identifying the rank positions of all the members of a defined gene set. To evaluate statistical differences, we used the signal-to-noise ratio as a statistic to compare specific and random phenotypes. Statistical significance was defined when FDR q-value < 0.25.

### 2.5. Gene Expression Data Analysis

Gene expression data from hepatoblastoma (HB) patients and non-tumoral tissue were obtained from both HTA (High-Throughput Assay) and RNAseq data previously published (11). Supervised analysis of gene expression data was performed using R software (v4.2.0) with the limma package, using a linear model with the empirical Bayes method. Heatmaps for gene expression visualisation were performed using the pheatmap package.

### 2.6. Immunohistochemistry (IHC)

The tissue sections were deparaffinised with xylene and rehydrated in IMS (denatured ethanol). Antigen retrieval was performed by incubating the slides in sodium citrate buffer (pH 6; Sigma-Aldrich, St. Louis, MO, USA) for 40 min in a steamer. The slides were then blocked first with 20% NGS (normal goat serum; Jackson ImmunoResearch, UK) in PBS for 5 min and later with Peroxide-Blocking Solution (Agilent, Santa Clara, CA, USA) for 5 min. Subsequently, the primary antibodies, CPT1a (Rabbit mAb; Cell Signaling Technology, Danvers, MA, USA) and CPT2 (Rabbit; Sigma-Aldrich, St. Louis, MO, USA), diluted in Antibody diluent (Agilent, Santa Clara, CA, USA) were applied on the positive control (duodenum) and test slides at 1:20 and 1:50 dilutions, respectively, and incubated for 1 h at room temperature. Antibody dilutions were previously optimised on duodenum as a positive control. Slides were then washed with PBS for 5 min and later incubated with the secondary antibody (Agilent, Santa Clara, CA, USA) for 30 min at room temperature. Slides were rinsed with PBS and incubated with DAB solution (Agilent, Santa Clara, CA, USA) for 5 min and rinsed with water. Afterwards, the specimens were counterstained with Hematoxylin Solution, Mayer’s (Sigma-Aldrich, St. Louis, MO, USA) for 30 s and then rinsed with water. The slides then underwent the series of IMS solution (from 95% to 100%) for dehydration and finished in xylene. The slides were mounted onto coverslips with DPX medium (Sigma-Aldrich, St. Louis, MO, USA) for further analysis. IHC slides were scanned using NanoZoomer (Hamamatsu Photonics K. K., Japan) at ×40 magnification and then viewed using the NDP.view2 software 2.9.29. Five to six random cores were selected per specimen and were saved at ×40 magnification in JPEG format. After image acquisition, ImageJ (Java 1.8.0_345; Wayne Rasband and contributors, National Institutes of Health, USA) analysis was performed. The threshold was optimised using the non-tumour tissues, and the chosen threshold was applied to all the IHC images, without any adjustment. There were at least three cores for each patient sample. The area fractions (%) were measured for each core using ImageJ; in the case of any disparity between core scores, the average score of protein level (area fraction) was recorded. IHC scoring data were analysed using Microsoft Excel and GraphPad Prism 9 and differences between non-tumour and tumour samples for CPT1a and CPT2 expressions were assessed by paired *t*-tests. The chi-squared test of independence was performed to look for any associations between clinicopathological variance and CPT1a and CPT2 expression. A value of *p* < 0.05 was considered statistically significant.

## 3. Results

### 3.1. The Metabolomic Profile of Hepatoblastoma Is Different to That of Non-Tumour Tissue in Paired Samples

Untargeted metabolomic profiling (Figure 1) shows that the profile of tumour tissue is different to that of paired non-tumour liver tissue taken from the same patient. The quality of the untargeted metabolomics data was checked using the QC and found to be good (89% of peaks had a coefficient of variation <30% in QC injections and QCs clustered in the centre of the principal component analysis (PCA plot)). There was one outlier, outside of the Hotelling’s T^2^ (corresponding to a multivariate generalisation of the 95% confidence interval); this was HB83_T (Figure 1A). The tumour and non-tumour samples showed some separation along PC1 with two tumour samples further to the right of PC1 (HB121_T and HB6_T) (Figure 1A). In supervised orthogonal partial least-squares discrimination analysis (OPLS-DA), the separation was clearer to see and HB6_T looked more like the non-tumour samples; the goodness of fit of the data to the model and the predictive ability of the model were good (R^2^Y 0.788, Q^2^ 0.481), and the permutation test validated the OPLS-DA model (Q^2^ y-axis intercept ≤ 0; Appendix A), so the separation was reliable (Figure 1B). HB83_T was still an outlier and was excluded from a further OPLS-DA plot (Appendix A). The metabolites responsible for the difference between groups were analysed using both multivariate (variable important for the projection (VIP) ≥1) and univariate (adjusted *p*-value <0.05) statistics. Metabolites which have differential abundance between HB and non-tumour tissue are listed in Table 1 and include amino acids, carnitines, metabolites in lipid anabolism/catabolism, three metabolites in glutathione metabolism, and TCA cycle metabolites. All were in lower abundance in tumour samples relative to non-tumour samples. Most metabolites were identified at a high level of confidence with fragmentation matches to an MS/MS database and retention time matches to a standard. The pathways that these metabolites are in were looked up in KEGG (https://www.kegg.jp/kegg/ (accessed on 11 July 2023)) and the Human Metabolome Database (https://hmdb.ca/ (accessed on 16 August 2023)). Metabolomic pathway analysis of the significantly altered metabolites was performed using MetaboAnalyst 5.0 (www.metaboanalyst.ca (accessed on 17 August 2023)). Based on the limit *p* < 0.05 (pathway enrichment) and pathway impact value > 0.10 (pathway topology) [24], alanine, aspartate and glutamate metabolism was a significant pathway (Appendix A). There is no carnitine pathway in KEGG, yet it was clear from our list of metabolites that the precursor to carnitine (4-trimethylammoniobutanoate), carnitine and five acylcarnitines were significantly altered in relative abundance in HB. Carnitine is used to transport fatty acids into and out of mitochondria.

### 3.2. Targeted Metabolomics Demonstrates a Reduction in Short-Chain Acylcarnitine Levels in Hepatoblastoma Tissue

Since acylcarnitines and amino acids were strongly represented in the list and their metabolisms interact, we used targeted metabolomics of acylcarnitines and amino acids to determine absolute concentrations (µmol per kg of dry tissue mass) in the same tissue pairs. This analysis confirmed that HB83_T is an outlier (Appendix A), so we excluded the pair from our statistics. Carnitine and short-chain (C2–C5) acylcarnitines, including, acetylcarnitine, propionylcarnitine, isobutrylcarnitine (from valine), 2-methylbutyroylcarnitine (from isoleucine), total of isovalerylcarnitine (from leucine) and 2-methylbutyroylcarnitine (from isoleucine), and hydroxybutyrylcarnitine were significantly less abundant in tumour compared to non-tumour tissue, whilst leucine (*p* = 0.078), isoleucine (*p* < 0.06) and valine (*p* < 0.06) trended towards being more abundant but did not achieve statistical significance (Figure 2). On the other hand, medium-chain (C6–C12) and long-chain (C13–C18) acylcarnitines were not significantly altered in abundance in cancer cells (Appendix A). The medium- and long-chain acylcarnitines arise from fatty acid oxidation, whereas the short-chain (especially branched) acylcarnitines may come from branched-chain amino acid catabolism. Succinylcarnitine-methylmalonylcarnitine was below the limit of quantification. Amongst other amino acids measured, alanine (as seen in the untargeted analysis; glutamine and glutamate were not significant in targeted metabolomics) and glycine were significantly decreased in abundance in tumours, whereas aspartate was significantly increased (Figure 2) and arginine trended towards an increase in tumour tissues (*p* < 0.06; Appendix A).

### 3.3. Metabolomic Profile in Hepatoblastoma Correlates with Transcriptomic Profile

In order to further investigate the dysregulated metabolic pathways in HB, we conducted a comparative study of gene expression between tumour (T) and non-tumour (NT) samples from patients with hepatoblastoma using previously published RNAseq and Human Tissue Array (HTA) data (11). Using RNAseq, we found that 49 out of 54 genes of interest (90.7%) related to the metabolic findings showed differential expression between HB and NT (Figure 3). It was noted that 39 out of 49 genes were downregulated in HB compared to NT (80%), and only 10 out of 49 (20%) were upregulated (*p* < 0.05). The results were confirmed using the HTA data, since we observed that 44 out of 53 genes of interest exhibited dysregulation between HB and NT (83%), and 43/44 (98%) were already found to be dysregulated using the RNAseq data (Figure 4). Similarly, 35 out of 44 genes (80%) were downregulated in HB compared to NT, and only 9 out of 44 were upregulated (20.5%) (*p* < 0.05). These genes were associated with metabolic pathways, such as fatty acid transport, glycolysis, the TCA cycle and branched-chain amino acid degradation, among others. The analysis of metabolic pathways using GSEA (gene set enrichment analysis) confirmed the dysregulation of these specific pathways (FDR < 0.25) in HB as compared to NT tissues (Appendix A). 

Transcriptomics showed downregulation of enzymes involved in the carnitine and branched-chain amino acid (BCAA) pathways, namely, carnitine palmitoyl transferase 1 (CPT1a), carnitine palmitoyl transferase 2 (CPT2), carnitine acylcarnitine translocase SLC25A20 (CACT), carnitine acetyltransferase (CrAT) and branched-chain alpha-keto acid dehydrogenase (BCKDH), and upregulation of fatty acid translocase (CD36), fatty acid binding protein 4 (FABP4) and branched-chain amino acid transaminase (BCAT) in tumour compared to non-tumour tissues. Interestingly, the downregulation of genes involved in the carnitine metabolic pathway in HB was further confirmed by gene set enrichment analysis (GSEA) (Figure 5).

### 3.4. CPT1a Is Downregulated in Hepatoblastoma

IHC was successfully conducted for both CPT1a and CPT2 expressions. Corroboratively, IHC showed lower expression of CPT1a in tumour (Figure 6A,B), showing downregulation of the carnitine cycle at both RNA and protein levels. There was no significant change found in tumour tissue compared to non-tumour tissue with regard to CPT2 expression. There was no significant association between the clinical and pathological features of paediatric patients with hepatoblastoma across gender, age, CHIC classification, PRETEXT and metastasis status. There was a significant association found between the change in CPT1a expression and histology types (histology confirmed in 12/15 cases). The chi-square test results indicated that the HB patients showing epithelial histology (*n* = 5) had an increase in CPT1a expression in tumour tissue compared to non-tumour tissue, while all samples showing mixed histology (*n* = 7) had a decrease in CPT1a expression in tumour tissue compared to non-tumour tissue (*p* < 0.05).

## 4. Discussion

To our knowledge, this paper is the first to use detailed high-resolution LC-MS/MS methods to delineate the metabolomic landscape of HB. Untargeted metabolomics using highly sensitive LC-MS/MS produced a list of 35 metabolites which were significantly altered in relative abundance between HB and paired healthy liver tissue. Many of these were inter-related within pathways, including catabolic pathways for glucose, fatty acids and branched-chain amino acids (Figure 7), which all have the potential to produce acylcarnitines of different lengths when catabolites do not quite reach the TCA cycle. Many of these metabolites are in pathways that converge in the mitochondria for energy production.

Collectively, the data in our study show a clear alteration of the fatty acid catabolism pathway in HB. CPT1a, which catalyses the rate-limiting step of fatty acyl-CoA transport from the cytosol into the mitochondria for subsequent oxidation [25], was downregulated in HB. In accordance with this, free carnitine, the primary substrate for CPT1, was also markedly less abundant in HB, with the cellular content being less than half of the NT content, suggesting that fatty acid oxidation could be dysregulated in HB. Transcriptomic analysis did, however, reveal an upregulation of both CD36 and FABP4, which are both involved in fatty acid transport into the cell; this may suggest that in HB, fatty acids are transported into the cell but not oxidised and may instead be diverted into other pathways (i.e., storage or anabolism). FABP1, which is the main liver isoform, was downregulated.

Free carnitine also performs another important metabolic role within the cell; it serves as a buffer of acetyl groups when acetyl-CoA is generated in excess of its utilisation by the TCA cycle [26]. Acetyl-CoA is formed from irreversible pyruvate oxidation, catalysed by the pyruvate dehydrogenase complex (PDC; Figure 7). To prevent the fall in the limited intracellular pool of free CoA and to prevent end-product (i.e., acetyl-CoA) inhibition of PDC, free carnitine accepts the acetyl group, thereby forming acetylcarnitine. The results of the current study demonstrate a clear reduction in acetylcarnitine content in HB and a reduction in CACT and CrAT (both key enzymes involved in acetylcarnitine formation) as well as PDC at the transcriptomic level, with untargeted metabolomics also revealing lower TCA metabolites in HB. We observed a lower level of glucose in HB in untargeted metabolomics, which agreed with Tasic et al., who also observed a higher lactate level [16], suggesting aerobic glycolysis. Taken together, these findings suggest a reduction in flux through PDC and the subsequent TCA cycle.

All three BCAAs showed a trend to be elevated in tumour liver samples. Interestingly, this elevation has been observed in several liver disease states, such as non-alcoholic liver disease and HCC [27]. Moreover, these findings have been observed at both the systemic and cellular levels. For example, elevated tissue BCAAs have been observed in patients with HCC previously [28,29], with the latter report also demonstrating a reduction in BCAA catabolic enzymes, in particular BCKDH, the rate-limiting enzyme that commits BCAA catabolites to their oxidative fate. In line with those observations, both BCKDH- and BCAA-derived short-chain acylcarnitines (which are formed downstream of the BCKDH step) were markedly quantitatively reduced in HB compared to healthy sections of liver in the present study. These findings point to a suppression of BCAA catabolism in HB, the clinical significance of which warrants further investigation. Given that reduction in pyruvate conversion to acetyl-CoA and reduction in fatty acid and BCAA catabolic pathways have been observed at the targeted and untargeted metabolomic and transcriptomic levels, it appears that HB can be collectively characterised as a state of reduced fatty acid and BCAA metabolism, with few molecules entering the TCA cycle. A downregulation of enzymes in endosomal ω- and peroxisomal and mitochondrial β-fatty acid oxidation, especially a severe deficiency of enoyl-CoA hydratase/3-hydroxyacylCoA dehydrogenase (EHHADH), was noted in HBs, and the Randle cycle appears to be at play in HB, where energy-generating metabolism switches from fatty acid oxidation (normally dominant in hepatocytes) to aerobic glycolysis (the Warburg effect) [30]. This was exploited by a diet of EHHADH substrate or product to extend survival by inducing necrosis in mouse models [30].

Our data largely agreed with Tasic et al.’s untargeted metabolomics of tumour and non-tumour pairs by HR-MAS, in which amino acids, including BCAA, lactate and fatty acids, were increased in abundance in tumours, whilst lipids, glutamine, glutamate and glucose were decreased in abundance [16]. Tasic et al.’s study also largely agreed with their group’s previous untargeted metabolomics of tumour and non-tumour non-paired samples, as alanine, phenylalanine, tyrosine, choline and formate were increased in abundance in both of their studies, whereas certain classes of lipids used as an alternative source of energy were decreased in abundance [15]. The main difference from our work was that alanine increased in abundance in HB in their studies, whereas it decreased in abundance in our study. Our study went further into understanding the HB metabolome, as we detected more metabolites involved in more pathways. Corroborating the Krepischi group’s work on NNMT downregulation at the transcriptional and protein levels in HB [15], our transcriptomics cohort also showed a decrease in NNMT expression in HB, and its substrate nicotinamide was decreased in abundance in HB, suggesting that production of this substrate was reduced in favour of more useful metabolites (Appendix A). NNMT consumes methyl donor groups in liver tissue, where S-adenosylmethionine (SAM) is the methyl donor (15), and SAM is produced from betaine (a methyl donor) in the methionine cycle; therefore, the lower abundances we saw of both betaine and nicotinamide in HB may be related. When SAM is consumed, there are fewer epigenetic marks; therefore, we surmise that HB could be an epigenetically driven cancer, as noted by the JCCG previously [12]. In fact, a strong epigenetic footprint has been reported in HB in a study of 32 matched pairs, where HB was characterised by genome-wide DNA hypomethylation similar to methylation patterns of foetal tissue [31], confirming the degree of immaturity of tumour cells in HB. Krepischi’s group hypothesised that NNMT downregulation might reduce HB lipid content, and we saw lowered glycerol backbones of lipids. Nicotinamide can also be used to produce NAD^+^, which is a major oxidising agent and used in catabolism to transfer energy to the electron transport chain through NAMPT and NMNAT2/3/1 enzymes (this salvage is the major route of NAD^+^ production). The fact that less nicotinamide seems to be produced and consumed may mean that energy production stops short of the TCA cycle and the electron transport chain, perhaps stopping at glycolysis. Another pathway which we saw to be affected in our untargeted analysis was the glutathione synthesis pathway, which was downregulated, with multiple genes lowered in expression and metabolites in lower abundance in HB (Appendix A). Glutathione is an antioxidant produced mainly by the liver, important in protecting the cell against oxidative stress, so the cancer cells reduce antioxidant abundance in order to rapidly proliferate.

Our study goes further than the previous studies by quantifying the main observed metabolomic changes in targeted analysis and confirming these at protein and gene expression levels. In particular, we found that carnitine and short-chain acylcarnitines were reduced in concentration in HB and that the carnitine metabolic pathway was dysregulated in the tumours (CPT1a, CPT2 and SLC25A20).

The main limitation of our study is that both the tumour and non-tumour samples used were obtained following neo-adjuvant chemotherapy rather than at diagnosis in line with most publications on HB biology [8,9,10,11]. In our study, we cannot rule out that some of the findings in the HB metabolome may be related to chemotherapy treatment. Therefore, the next steps would be to validate the findings in this study in pre-chemotherapy biopsy specimens and in a larger number of samples. We will also need to elucidate the underlying mechanisms driving these metabolomic changes and examine the role that the underlying aberrant pathways play in modulating cell survival in HB. It is possible that this future work might lead to the development of novel therapies.

## 5. Conclusions

In summary, we gained new insights into the metabolomic landscape of HB and detected many differentially abundant metabolites using our sensitive LC-MS/MS method, including many acylcarnitines, previously unreported in HB. Targeted metabolomics of acylcarnitines strengthened our untargeted metabolomics findings, which were further validated at the protein and gene expression levels. 

## Figures and Tables

**Figure 1 cancers-15-05182-f001:**
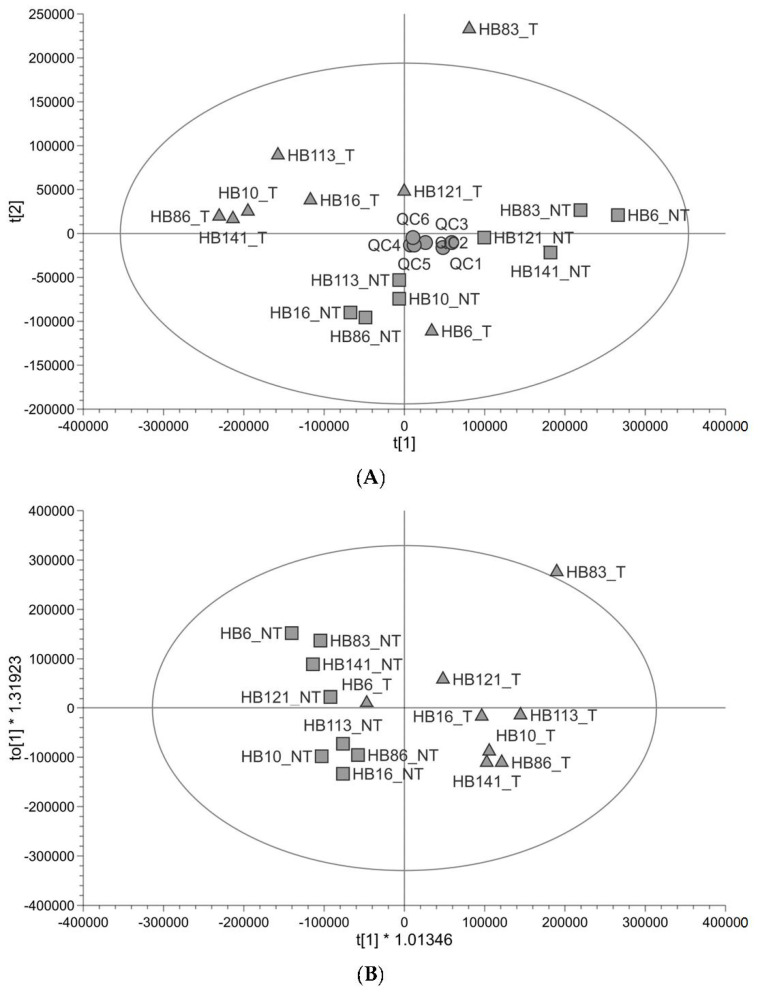
Untargeted metabolite profiling in paired tumour and non-tumour tissue samples (*n* = 8). (**A**) PCA score plot showing hepatoblastoma (▲) and healthy liver (■) paired samples from paediatric patients alongside pooled QCs (●). (**B**) OPLS-DA score plot showing the same samples (R^2^Y 0.788, Q^2^ 0.481).

**Figure 2 cancers-15-05182-f002:**
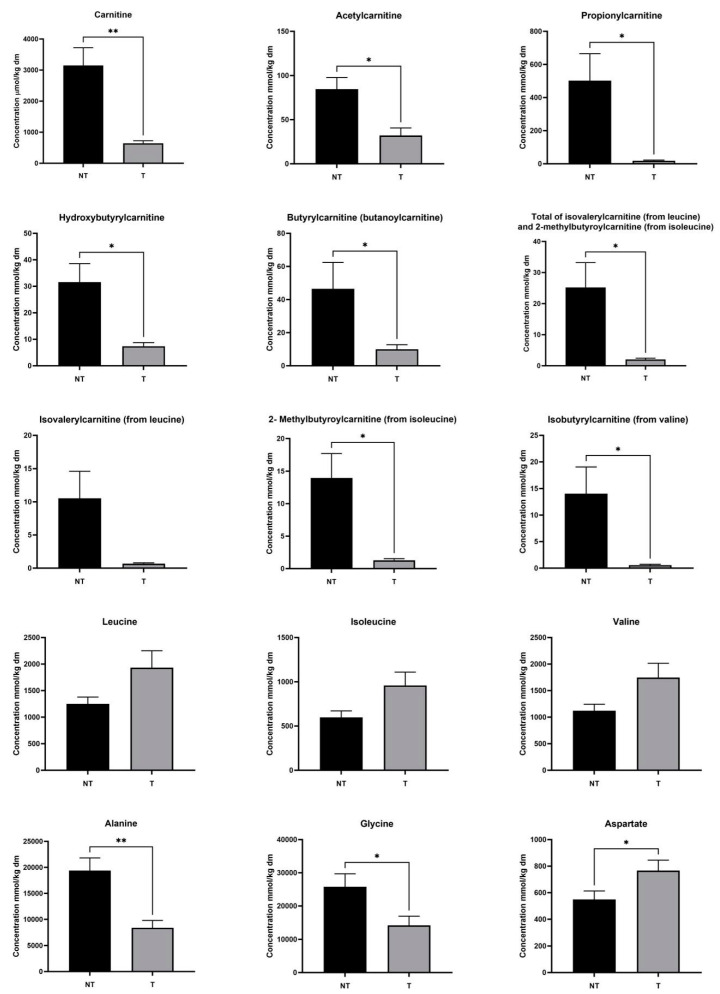
Amino acid, free carnitine and acylcarnitine concentrations in hepatoblastoma (T) and paired healthy liver tissue (NT). Values are means ± standard deviations; *n* = 7 patients; * *p* < 0.05, ** *p* < 0.01.

**Figure 3 cancers-15-05182-f003:**
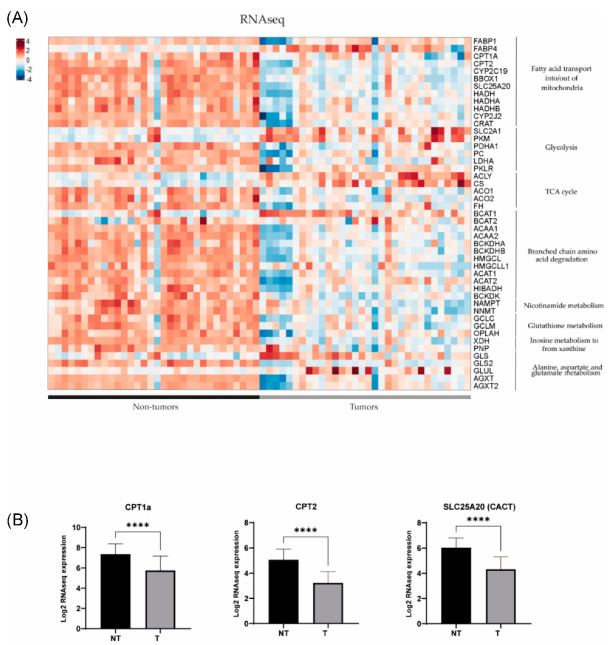
(**A**) Gene expression differences assessed by RNAseq in key metabolic pathways in 32 tumour (T) and 32 non-tumour (NT) samples (data extracted from Carrillo-Reixach et al. [11]). (**B**) Differences in the expression of CPT1a, CPT2 and SLC25A20 (CACT) in tumours in comparison with non-tumours. *p*-values were calculated using unpaired *t*-tests. **** *p* < 0.0001.

**Figure 4 cancers-15-05182-f004:**
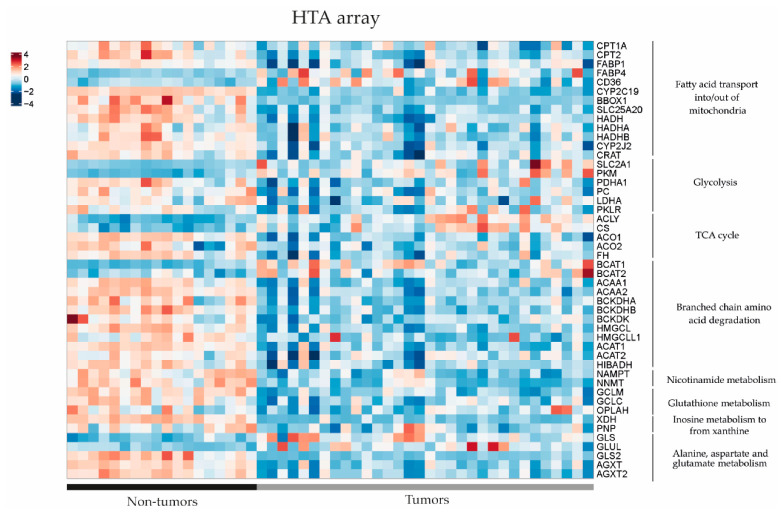
Gene expression differences in the key metabolic pathways assessed by Human Transcriptome Array (HTA) heatmap in 32 tumour (T) and 18 non-tumour (NT) samples (data extracted from Carrillo-Reixach et al. [11]).

**Figure 5 cancers-15-05182-f005:**
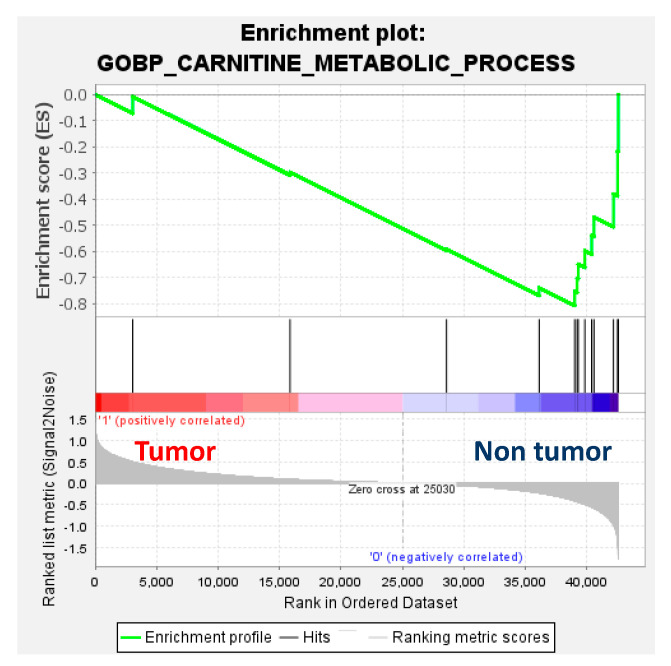
Gene set enrichment analysis (GSEA) enrichment plot (normalised enrichment score = −1.56; *p* = 0.16; false-detection rate (FDR) q-value = 0.16) using the RNAseq dataset of 32 paired tumour and non-tumour samples from Carrillo-Reixach et al. [11] and the gene set of the Gene Ontology Biological Process (GOBP) of the carnitine metabolic process of the Molecular Signatures Database (MSigDB).

**Figure 6 cancers-15-05182-f006:**
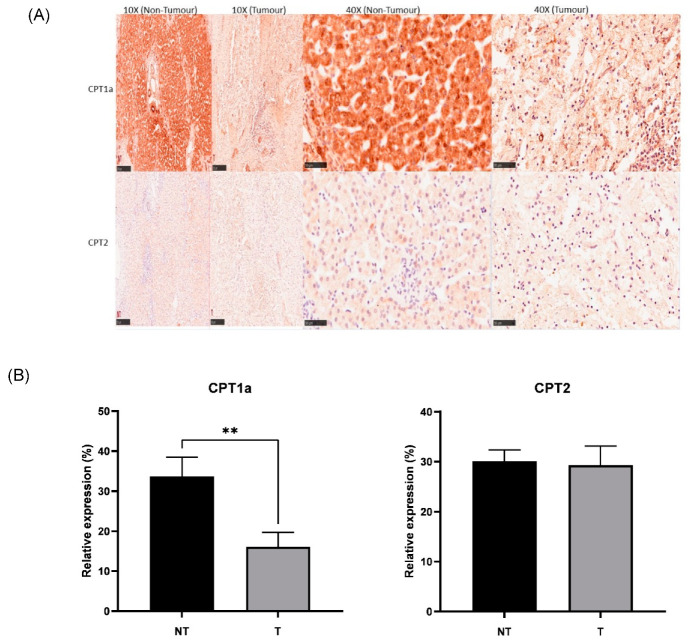
(**A**) Representative IHC staining of CPT1a and CPT2 in the HB (tumour) and NT (non-tumour) regions (scale bar: 250 μm for 10× and 50 μm for 40×). (**B**) Bar graph showing expression of CPT1a and CPT2 in hepatoblastoma (T) and paired healthy liver tissue (NT) areas. ** *p* < 0.01. Five random areas from each tissue were quantified (means ± SEMs).

**Figure 7 cancers-15-05182-f007:**
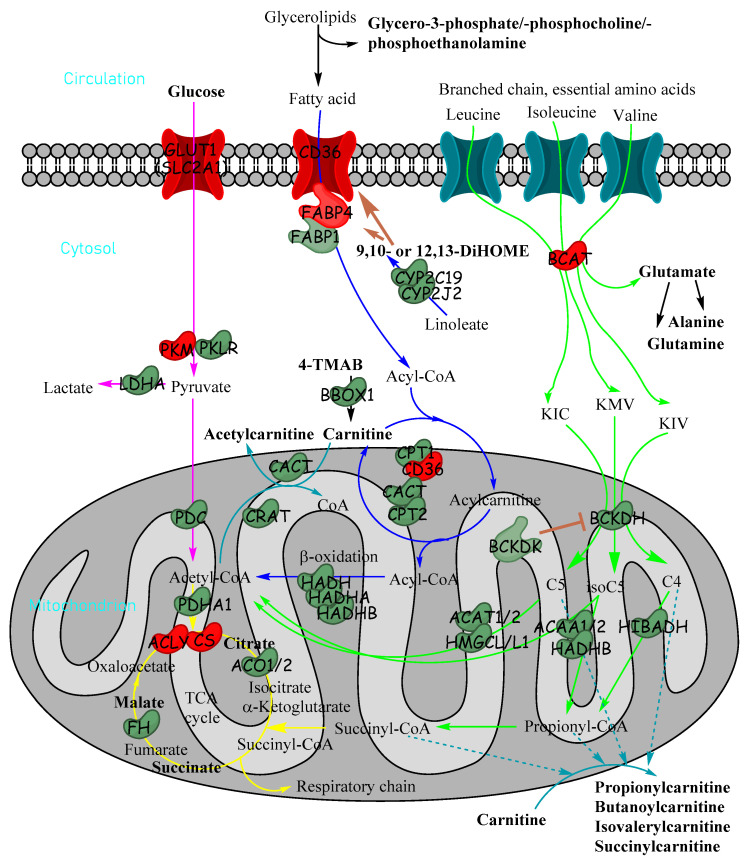
Metabolic pathways in cells showing how glucose, fatty acids and essential branched-chain amino acids are utilised for energy production in cells and how acylcarnitines can be formed from fatty acids and branched-chain amino acids. Metabolites in bold were significantly reduced in untargeted metabolomics in tumour tissue relative to non-tumour tissue. Enzymes and transporter proteins and kinases shown are significantly altered in expression (transcriptomics) in HB (red indicates increase; green indicates decrease): α-ketoisocaproate (KIC), α-keto-β-methylvalerate (KMV), α-ketoisovalerate (KIV), 4-trimethylammoniobutanoate (4-TMAB), isovaleryl-CoA (C5), α-methylbutyryl-CoA (isoC5) and isobutyryl-CoA (C4/isoC4).

**Table 1 cancers-15-05182-t001:** Differentially abundant metabolites mapping onto metabolic pathways (* refer to methods section).

Annotation	Pathway	Fold Change (Tumour)/(Non-Tumour)	FDR-Corrected *p*-Value	VIP Value	Level of Identification *
4-Trimethylammoniobutanoate	Fatty acid transport into/out of mitochondria	0.57	6.53 × 10^−3^	1.3	2
L-Carnitine	Fatty acid transport into/out of mitochondria	0.21	3.66 × 10^−5^	12.3	2
O-Acetylcarnitine	Fatty acid transport into/out of mitochondria	0.30	6.58 × 10^−5^	4.8	2
Propionylcarnitine	Fatty acid transport into/out of mitochondria	0.07	5.97 × 10^−6^	4.5	2
O-Butanoylcarnitine	Fatty acid transport into/out of mitochondria	0.54	1.48 × 10^−3^	1.4	2
Isovalerylcarnitine	Fatty acid transport into/out of mitochondria	0.14	6.58 × 10^−5^	1.1	3
O-Succinylcarnitine	Fatty acid transport into/out of mitochondria	0.07	3.70 × 10^−7^	1.2	4
9,10-DiHOME	Cell signalling/fatty acid beta-oxidation	0.03	1.58 × 10^−7^	1.0	4
1-Palmitoylglycerophosphocholine	Glycerophospholipid metabolism	0.34	3.89 × 10^−5^	1.6	4
sn-Glycero-3-Phosphocholine	Glycerophospholipid metabolism	0.07	3.63 × 10^−9^	6.2	2
sn-Glycero-3-phosphoethanolamine	Glycerophospholipid metabolism	0.04	2.66 × 10^−7^	1.9	4
sn-Glycerol 3-phosphate	Glycerophospholipid metabolism	0.18	7.38 × 10^−6^	1.5	2
4-Oxoproline	Unknown	0.62	2.10 × 10^−3^	2.8	4
5-Oxoproline	Glutathione metabolism	0.47	4.40 × 10^−5^	1.7	2
L-Glutathione (reduced)	Glutathione metabolism (an antioxidant)	0.05	3.01 × 10^−6^	1.8	3
L-Glutamate	Glutathione metabolism; alanine, aspartate and glutamate metabolism	0.89	3.39 × 10^−2^	1.2	2
L-Glutamine	Alanine, aspartate and glutamate metabolism	0.53	8.86 × 10^−5^	1.1	2
L-Alanine	Alanine, aspartate and glutamate metabolism	0.31	5.22 × 10^−5^	1.6	2
Creatine	Facilitates recycling of ATP; arginine and proline metabolism	0.29	2.48 × 10^−6^	11.5	2
D-Glucose	Glycolysis	0.47	1.66 × 10^−4^	2.4	2
Citric acid	TCA cycle	0.59	5.94 × 10^−3^	1.9	2
Succinate	TCA cycle	0.68	4.24 × 10^−3^	2.1	2
(S)-Malate	TCA cycle	0.28	2.66 × 10^−4^	3.0	2
Nicotinamide	The main source of NAD + (which is a major oxidising agent)	0.45	1.39 × 10^−4^	3.9	2
Betaine	One-carbon metabolism	0.36	1.21 × 10^−4^	11.4	2
Inosine	Nucleotide metabolism (purine)	0.67	4.37 × 10^−4^	1.6	2
Xanthine	Nucleotide metabolism (purine)	0.30	7.42 × 10^−7^	1.6	2
Uridine	Nucleotide metabolism (pyrimidine)	0.46	1.51 × 10^−4^	1.2	2
L-2-Aminoadipate	Lysine degradation	0.60	2.55 × 10^−2^	1.3	3
Norecasantalic acid	Unknown	0.02	9.72 × 10^−8^	1.2	4
3-Acetamidopropanal	Unknown	0.34	9.38 × 10^−5^	1.3	4
D-Erythrose	Carbohydrate	0.25	1.18 × 10^−5^	1.1	4
Diazoxide	A drug	0.04	1.44 × 10^−5^	3.1	3
Dimethyl maleate	Unknown	0.41	9.08 × 10^−5^	1.0	4
Dimethyl maleate	Unknown	0.60	2.55 × 10^−2^	1.3	4

## Data Availability

Processed metabolomics data (peak areas) can be found in the Excel files in the Appendix A. Raw data files and IHC images can be requested from the authors. Transcriptomic data were previously published.

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
