# Peer review of "Characterisation of Aberrant Metabolic Pathways in Hepatoblastoma Using Liquid Chromatography and Tandem Mass Spectrometry (LC-MS/MS)"

_cancers, 2023, doi:10.3390/cancers15215182_

Round 1
Reviewer 1 Report
Comments and Suggestions for Authors
In this manuscript, the authors conducted a study on hepatoblastoma (HB), a rare childhood tumor, with a focus on its metabolomic characteristics. The authors employed liquid chromatography coupled with mass spectrometry to analyze samples from eight HB patients and compared them to non-tumor liver tissue. They discovered that HB exhibits a distinct metabolomic profile, characterized by changes in pathways related to fatty acid transport, glycolysis, the TCA cycle, branched chain amino acid degradation, and glutathione synthesis. The authors also revealed reduced levels of certain acylcarnitines and an accumulation of branched chain amino acids in HB. These findings were consistent with previous data obtained through different methods. Gene expression analysis indicated a decrease in the carnitine metabolome, suggesting a reduced utilization of long-chain fatty acids in HB. This research is intriguing and could potentially lead to new therapeutic targets for HB treatment; therefore, I recommend accepting this manuscript.
Author Response
We thank reviewer 1 for taking the time and effort to review our article and for the favourable opinion they have provided. As they haven't suggested any changes, we have not attached any additional information for this response. We have taken the editors comments and that of reviewer 2 on board and made the suggested changes as recommended.
Reviewer 2 Report
Comments and Suggestions for Authors
The manuscript is more complex than it tiltles reads out. Each secion need to simplified in order to make it understable. I have a few suggestions which need to be included before publication.
1) It would be great if the author explains in detail about the metabolic pathway for hepatoblastoma in the introduction. 2) It would be interesting to see the chromatogram for targeted and non targeted metabolites? which will make the readers understand well. 3) What does the table 1 represent? It will need better explanation in terms of each metabolites associated with the pathways mentioned in the table 1. 4) Explain the importance of each of the metabolites and its results in the discussion secession rather just submitting bunch fo supplementary documents which hard to read and understand what authors are trying to achieve.Author Response
Thank you for reviewing our paper and for suggesting constructive changes. We have taken all your comments on board and have responded to your questions as follows
Q1) It would be great if the author explains in detail about the metabolic pathway for hepatoblastoma in the introduction.
There is very little information known about metabolic pathways in hepatoblastoma. Our paper is the first one to employ LC-MS/MS based methods to delineate hepatoblastoma metabolome. The only other group ( Tasic et al) employed HR- MAS methodology which we have cited and mentioned in the introduction and discussion.
2) It would be interesting to see the chromatogram for targeted and non targeted metabolites? which will make the readers understand well.
We have provided this chromatograms as you have suggested ( attached file below).
3) What does the table 1 represent? It will need better explanation in terms of each metabolites associated with the pathways mentioned in the table 1.
We have re-done table 1 to make the information much clearer and added to the results section.
4) Explain the importance of each of the metabolites and its results in the discussion sections rather just submitting bunch of supplementary documents which hard to read and understand what authors are trying to achieve.
This has been covered in the discussion; however, based on feedback we have expanded the results and discussion section to explain the importance of the significant metabolites
